# Diagnosis of Canine Tumours and the Value of Combined Detection of VEGF, P53, SF and NLRP3 for the Early Diagnosis of Canine Mammary Carcinoma

**DOI:** 10.3390/ani14091272

**Published:** 2024-04-24

**Authors:** Ning-Yu Yang, Hui-Hua Zheng, Chao Yu, Yan Ye, Guang-Hong Xie

**Affiliations:** 1College of Veterinary Medicine, Jilin University, No. 5333 Xi’an Road, Changchun 130062, China; yangningyu0925@163.com (N.-Y.Y.); zhh@zafu.edu.cn (H.-H.Z.); yc939462630@163.com (C.Y.); 13351590278@163.com (Y.Y.); 2College of Animal Science and Technology & College of Veterinary Medicine, Zhejiang A&F University, Hangzhou 311300, China

**Keywords:** canine tumours, tumour markers, diagnosis, combined tests

## Abstract

**Simple Summary:**

In recent years, canine tumours, as a common clinical disease with high lethality, have gradually received public attention. At present, visual diagnosis, X-ray and cytological examination are commonly used for clinical diagnosis, but at this time, the tumour is mostly diagnosed in the middle and late stages, and the therapeutic effect is poor. Tumour marker tests are not widely used, but are of great significance in the aspects of early diagnosis and prognosis assessment. In this study, we observed the metastasis of the tumour in dogs by imaging, understood the body condition of the animals by haematological examination, initially determined the type of the tumour by cytological examination, and determined the nature of the tumour by pathological histological examination. Then, we detected the high expression of tumour markers in the canine tissues and serum with mammary carcinoma by RT-qPCR and ELISA. The difference was highly significant between the benign and the control groups (*p* < 0.001). The combined detection of the four markers was found to achieve the highest confirmatory rate by ROC curves.

**Abstract:**

The average life of a dog is generally maintained at ten to fifteen years, and tumours are the predominant reason that leads to the death of dogs, especially canine mammary carcinoma. Therefore, early diagnosis of tumours is very important. In this study, tumor size, morphology, and texture could be seen through general clinical examination, tumor metastasis could be seen through imaging examination, inflammatory reactions could be seen through hematological examination, and abnormal cell morphology could be seen through cytological and histopathological examination. In the 269 malignant cases and 179 benign cases, we randomly selected 30 cases each, and an additional 30 healthy dogs were selected for the experiment (healthy dogs: dogs in good physical condition without any tumor or other diseases). We used RT-qPCR and ELISA to determine the relative expression of vascular endothelial growth factor (VEGF), tumor protein P53 (P53), serum ferritin (SF), and NOD-like receptor protein 3 (NLRP3) in 30 healthy dogs, 30 dogs with benign mammary tumours, and 30 dogs with malignant mammary tumours. In the results, the same expression trend was obtained both in serum and tissues, and the expression of the four markers was the highest in malignant mammary tumours, with highly significant differences compared with the benign and healthy/paracancerous groups. By plotting the ROC curves, it was found that the results of combined tests were better than a single test and the combination of the four markers was the best for the early diagnosis. In conclusion, this can assist the clinical early diagnosis to a certain extent, and also provides some references and assistance for the development of tumor detection kits in clinical practice.

## 1. Introduction

In clinical practice, tumour is the most common disease in dogs and is classified into benign and malignant tumours [1]. Knowing the type and nature of the tumor is particularly important [2]. Currently, common clinical diagnostic methods include general examinations such as visual inspection and palpation, imaging examinations such as X-ray, computed tomography (CT) and ultrasonography, haematological examinations such as blood routine and biochemistry, cytological examination, pathological histological examination, etc. [3]. Different diagnostic methods are selected according to the location and nature of the tumours. 

Tumour markers are a class of substances that are synthesized by the tumour cells themselves, or are released or produced at elevated levels by the body in response to the tumour cells [4]. Tumour markers exist in cells, tissues, blood, and body fluids, and are mainly used for auxiliary diagnosis, prognostic judgment, therapeutic effect observation, and monitoring recurrence, and can be used as one of the indicators in an early clinic diagnosis of tumour [5].

As one of the hotspots in inflammatory diseases and medical oncology research in recent years, the activated NLRP3 inflammasome regulates the tumor microenvironment by controlling the secretion of Caspase~1, IL-1B, and IL-18, which induces oxidative DNA damage and uncontrolled proliferation of tissue cells, and promotes the formation and development of tumours [6]. It also promotes the growth and metastasis of human oral squamous epithelial, and is negatively correlated with the clinical staging and pathologic grading of liver cancer [7]. These inconsistent results suggest that the NLRP3 inflammasome may play a role in promoting or inhibiting tumorigenesis in different neoplastic diseases [8]. 

P53 gene abnormality is one of the most common genetic damages in mammary carcinoma, and the P53 gene product is overexpressed in canine mammary tumours, which can be used as one of the major prognostic markers [9,10]. Since the first discovery of P53, more and more studies on P53 have been conducted; mutations in the P53 gene have also been shown to be closely associated with tumorigenesis [11,12]. In canine mammary tumours, there are relatively few reports on P53 gene mutations, especially on the detection of the entire sequence of the coding region of the P53 gene [13]. The early literature showed that VEGF was a cytokine that promoted the proliferation of vascular endothelial cells and induced the differentiation and proliferation of tumor cells, and its level was related to the size of the tumor, lymph node metastasis, and the stage of the tumor, etc. [14]. Its expression is high in mammary carcinoma and as one of the important biological indexes, it can be used to assess the progression of mammary carcinoma [15]. 

Some researchers also proposed that if tumours would like to generate blood vessels, they would secrete some angiogenesis-inducing factors and new blood vessels would be established when there was insufficient oxygen in the tumor [16]. In other words, the presence of tumours could induce angiogenesis. Tumor growth is controlled by the surrounding blood vessels, so inhibition of angiogenesis could lead to tumor dormancy, and the growth of new blood vessels promotes tumor growth, leading to the discovery of vascular endothelial growth factor [17,18]. There is a study that has shown that VEGF and its receptors can be highly expressed in tumor-affected dogs, but they are not related to overall survival [19]. 

In recent years, more and more studies have found that SF has a complex and subtle relationship with tumorigenesis, progression, and treatment, and it has a double-edged sword effect by inhibiting tumor growth as well as promoting tumor progression [20,21]. In tumor diagnosis, SF can be used as an important biomarker to assess tumor occurrence, development and prognosis [22].

In this study, we first used common clinical examinations to understand the basic conditions of the affected dogs, and then used HE staining to determine the nature of the tumours. The expression of the four markers in the tissues was probed by RT-qPCR, and the trend was the same by ELISA, so as to analyse the significance of the combined tests and to provide help for making a clinical diagnosis.

## 2. Materials and Methods

### 2.1. Animals and Sampling

From 2021 to 2023, information on 448 tumour-affected canine cases was collected from 11 pet hospitals in Changchun, Jilin Province, China. The data of the cases included breed, age, sex, spayed status, somatotype, tumor location and dietary habits. Then, 30 benign mammary tumour tissues and sera, 30 malignant mammary tumour tissues and sera, 30 benign mammary tumour paracancerous tissues, and 30 healthy canine sera were selected from among these cases. Paracancerous tissues were sampled 1 cm from the centre of the tumour tissue. After surgical removal, the tumour tissues and paracancerous tissues were placed in 1.5 mL sterile, enzyme-free tubes and stored in a −80 °C freezer together with the serum samples. Blood routine examinations and biochemical examinations were conducted within two hours.

### 2.2. Clinical Tumor Examinations

By asking the owners, we learned about the dogs’ basic information including sex, age, breed, dietary habits, reproductive history, sterilization history, past medical history, living environment, disease time, clinical symptoms and abnormalities in daily life. We also observed the site, size, colour and shape of the swelling as well as the dogs’ state through visual examination. Through palpation, we determined whether the swelling was free and made sure of its texture.

### 2.3. Imaging Examination

X-ray (AV choice 400 Plus, DEL MEDICAL, Bloomingdale, Naperville, IL, USA) was used to check for the metastases of the tumours. Appropriate parameters were selected based on the dogs’ body size, and the centre of the projection cross was aligned with the area to be examined. In order to ensure the accuracy of the results, it is usually recommended to perform the procedure twice, once in the orthostatic position and the other in the lateral position [23]. Ultrasonography (MyLab™ Six VET, Esaote, Genoa, Italy) is mainly used to diagnose tumours in soft tissues by using the echo differences formed by different tissues and sweeping through different sides. At the same time, ultrasonography can be used to observe whether adhesion occurs between the swellings and the surrounding tissues as well as the blood flow and size [24]. 

The imaging principle of CT (SIEMENS MAGNETOM ESSENZA Galaxy, Berlin, Germany) is similar to that of X-ray, but errors caused by posing problems can be well avoided. The shape, location, boundary, metastasis and invasion of the tumour can be understood through CT [25]. However, this method is not very common in clinical practice since the cost of the instrument and technology is very high. In addition, due to the time-intensive nature of CT scanning, it is easily affected by gastrointestinal peristalsis, respiration and other physiological activities, resulting in artefacts and missed diagnosis.

### 2.4. Haematology Examination

Blood routine examinations: For dogs in good physical condition, blood was drawn from the veins of the forelimbs or hindlimbs into a prepared EDTA anticoagulation tube, and for dogs in poorer physical condition or smaller dogs, blood was drawn from the jugular vein. The EDTA anticoagulant tube was shaken up and down slowly with one hand, and the blood was analysed using a fully automated blood routine analyser.

Blood biochemical examination: Blood from the forelimb, hindlimb or jugular vein was drawn into the heparin tubes and centrifuged at 5000 r/min in a high-speed centrifuge for 5 min. The supernatant was aspirated with a pipette, and then tested using a fully automatic biochemistry instrument.

### 2.5. Cytological Examination

Prior to cytologic specimen preparation, six or more slides were placed in a surgical instrument tray and the slide surface was wiped clean with a towel until the surface was free of glass debris, etc. The sampling site was properly cleaned and sterilized first. Then the tip of the needle was inserted into the mass to be examined and the piston was withdrawn to generate negative pressure. The mass was punctured several times in different directions, then the negative pressure of the syringe was released and the needle was withdrawn [26]. Thus, the empty needle sucked in air and was held steadily in the left hand. The needle hole was bevelled downward. Then the piston of the syringe was pushed quickly and the aspirated tissue particles and mucus were ejected on the slide. The smears were dried with an alcohol lamp flame and fixed with Diff-Quik Fixative for 20 s. The smears were first stained with Diff-Quik I for 5–10 s (lifting the slide up and down 2–3 times for distribution) and then taken out immediately after staining with Diff-Quik II for 10–20 s (lifting the slide up and down 2–3 times for distribution). It was rinsed with running water and dried naturally, placed under a microscope for observation, first under low magnification, and then observed with a high magnification oil microscope.

### 2.6. Pathology Examination

The excised fresh tumor tissue was stained after fixation, washing and dehydrated. The morphology and structure of the tumor cells were observed under the microscope with different magnifications to determine the benign or malignant nature of the tumor. If the tumor was malignant, it would be graded according to the tumor tissue grading system.

### 2.7. Molecular Biological Detection

#### 2.7.1. Primer Design

The sequences of canine GAPDH, VEGF, SF and P53 genes were queried on the NCBI website (Table 1), and the specific primers were designed by Primer Premier 5.0 (DNASTAR Inc., Madison, WI, USA). GAPDH was selected as the reference gene. All of the primers were synthesized by Jilin Kumi Biotechnology Co., Changchun, China.

#### 2.7.2. Agarose Gel Electrophoresis

The extracted total RNA was reverse transcribed in a 20 µL reaction volume including 4 µL 5 × SweScript All-in-One SuperMix for qPCR, 1 µL gDNA Remover, 0.1 ng–5 µg/10 pg–0.5 µg Total RNA, nuclease-free water, and the program was set to 25 °C for 5 min, 42 °C for 20 min, and 85 °C for 5 s. The conventional PCR amplification was performed using a PCT-200 Peltier Thermal Cycler (MT Research, Waltham, MA, USA). The amplification was carried out in a 25 µL reaction volume containing 12.5 µL of 2 × Taq PCR Master Mix (P112-01, Vazyme, Nanjing, China), 3 µL of cDNA template, 0.5 µL of each primer, and 8.5 µL of deionized water. The PCR parameters were 94 °C for 2 min; followed by 35 cycles of 94 °C for 30 s, 58 °C for 30 s, and 72 °C for 15 s, with a final extension at 72 °C for 10 min. The amplification products were visualized by 1.0% agarose gel electrophoresis (1645052, 1704486, Bio-Rad, Hercules, CA, USA).

#### 2.7.3. Real-Time PCR

Real-time PCR amplification of the four markers was performed using an Mx3005P-qPCR system (Agilent Technologies, Santa Clara, CA, USA). The amplification volume was 20 µL, including 10 µL 2 × Universal Blue SYBR Green qPCR Master Mix, 0.4 µL forward primer (10 M)^a^, 0.4 µl reverse primer (10 M)^a^, 2 µL template, and 7.2 µL nuclease-free water. Real-time PCR parameters were 95 °C for 30 s, followed by 40 cycles of 95 °C for 15 s, 60 °C for 10 s, and 72 °C for 30 s, followed by the instrument’s default melting process. GAPDH was used as an internal reference and three replicate wells were made for each sample of the four markers to minimize errors. The results including CT and so on were obtained by the Mx3000/Mx3005P real-time PCR system (Shanghai, China) and the 2^−∆∆Ct^ method was used for calculation and to compare the differences in the mRNA expression of the target genes in a relative quantitative method. The results were processed and analysed by Graphpad Prism 9.0 statistical software, and the comparisons between different groups were analysed by the *t*-test. The differences were considered significant and statistically significant when the *p*-values were less than 0.05.

#### 2.7.4. ELISA

The levels of the four markers in serum were assayed using ELISA kits, including canine P53 ELISA kit, canine NLRP3 ELISA kit, canine SF ELISA kit and canine VEGF ELISA kit. The required plates were removed from the aluminium foil pouch after equilibrating at room temperature for 20 min, and the remaining plates were sealed with a self-sealing bag, then put them back to 4 °C. Standard and sample wells were set up. First, 50 µL of different concentrations of standards were added to each of the standard wells, then 10 µL of the samples to be tested were added to the sample wells, followed by 40 µL of the sample diluent. Nothing was added to the blank well. Each well received 100 μL of HRP-conjugate reagent, covered with an adhesive strip and incubated for 60 min at 37 °C. Each well was aspirated and washed, repeating the process four times for a total of five washes. Each well was washed with Wash Solution (400 μL) using a squirt bottle, manifold dispenser or autowasher. Complete removal of the liquid at each step is essential for good performance. After the last wash, any remaining wash solution was removed by aspirating or decanting. The plate was inverted and blotted against clean paper towels. Chromogen solution A 50 μL and chromogen solution B 50 μL were added to each well. After gently mixing it was incubated for 15 min at 37 °C. Each well then received 50 μL stop solution. The colour of the wells should change from blue to yellow. If the colour in the wells was green or the colour change did not appear uniform, the plate was gently tapped to ensure thorough mixing. The optical density (O.D.) at 450 nm was read using a microtiter plate reader within 15 min. Then, a standard curve was drawn.

## 3. Results

### 3.1. Clinical Tumor Examinations

In the pictures of clinical manifestations of dogs with different tumours, distinctly raised masses can be seen (As shown in the red box) and either firmly or fluctuatingly palpated. The basic features are shown in Figure 1. In order to more intuitively compare the clinical characteristics between diseased and healthy dogs, we summarized them in Table 2.

### 3.2. Imaging Examination

Figure 2A–J include the tumor morphology by ultrasound, X-ray, and CT imaging and the tumours can be seen in the red boxes.

The mass is heterogeneous in texture and may be malignant if it appears as a hyperdense image (Figure 2A–C).

The tumour site in the affected dogs showed either higher or lower-density images than the surrounding tissues (Figure 2D–F).

The results of the CT examination showed that the three-dimensional images are able to better present the size, morphology and relationship of the tumour to the surrounding tissues (Figure 2G–J).

### 3.3. Haematology Examination

The blood counts of dogs with mammary tumours were analysed by routine blood tests (Table 3), and it was found that the number of inflammatory cells such as leukocytes were elevated in individual dogs with tumours compared to healthy dogs, suggesting that there is a certain inflammatory response in the organism. Decreased levels of lymphocytes suggest that there may be damage to the immune system or a septic infection, which is usually seen in the routine blood reports of dogs with malignant tumours, and which may be clinically manifested by tumour breakdown and haemorrhage. Table 4 shows the blood biochemistry results of dogs with mammary carcinoma, which have high alanine transferase values, suggesting possible liver problems, and extraordinarily high alkaline phosphatase values, suggesting hepatobiliary problems.

### 3.4. Cytological Examination

Figure 3A,D,G,J show the clinical characterization of the tumors, while the remaining images show the microscopic features of the tumors. The tumor site is marked with red circle, and abnormal areas under the microscope are indicated with red arrows.

As shown in Figure 3, Figure 3A is a picture of the clinical signs of oral melanoma; in Figure 3B,C, metastatic melanocytes, in addition to lymphocytes, are present, suggesting malignancy of the tumour.

Figure 3D shows a picture of clinical symptoms in an affected dog; Figure 3E,F show a large number of round cells with homogeneous vacuoles in the cytoplasm, nuclei to one side, basophilic pale staining of the cytoplasm, some cells with obvious nucleoli of varying sizes, vacuoles and a large number of erythrocytes in the background, in which mitosis can also be seen.

Figure 3G shows the site of the tumour; Figure 3H,I show a higher number of cells, deeper staining of the nuclei, and the presence of a large number of homogeneous lymphocytes, which is suspected to be a possible lymphoma because the tumour is located in a non-lymph node site on the body surface.

Figure 3J is a picture of clinical symptoms in a dog with a mammary tumour; in Figure 3K,L, in addition to the presence of epithelial cells, a large number of mesenchymal cells are seen and the tumour is growing in the mammary area, so it is considered that it might be ectopic osteosarcoma of the mammary gland. The cell borders are blurred, a large number of cell clusters are seen, and the intercellular adhesion is strong and tightly arranged.

### 3.5. Pathology Examination

The results of pathological histological sections of canine tumours are presented in Figure 4. The red arrows in the picture indicate the abnormal location.

Figure 4A–C show canine mammary tumours. Under high magnification, there is chondrogenesis within the mass (Figure 4A, arrow pointing), and a large number of hyperplastic glandular epithelium and myoepithelium are seen; the hyperplastic glandular epithelial cells are cuboidal to columnar, interconnected to form luminal or nested or lamellar forms (Figure 4B, arrow pointing), with a high nuclear-to-cytoplasmic ratio, occasional karyokinesis, and a moderate degree of anisotropy (Figure 4C, arrow pointing), and a high degree of anisotropy (Figure 4D–F, arrow pointing).

Figure 4D–F show canine lipomas. Under the microscope, the mass is seen to have an incomplete capsule, with a large number of proliferating adipocytes infiltrating into the transverse muscle fibres, which are separated into lobules by the muscle fibres and connective tissues, forming a marble-like appearance (Figure 4D,F, arrow pointing). The adipocytes are well differentiated, the cytoplasm is vacuolated, and the nuclei are extruded and distorted at the cell margins (Figure 4E, arrow pointing).

Figure 4G–I show a canine mast cell tumour. Under high magnification, the tumour cells and their nuclei in the ulcerated area are spindle-shaped (Figure 4G, arrow pointing); the interior of the mass is filled with hyperplastic cells, nearly round, with abundant cytoplasm and clear demarcation; the nuclei are round or oval, located in the centre, with one or two obvious nucleoli, and occasional binucleated cells are seen; cytokinesis is basically invisible, and the cells are well differentiated, with a low anisotropy and a “Holland egg-like” appearance (Figure 4H, arrow pointing). The cells are well differentiated with low heterogeneity, resembling a “ruffled egg-like” appearance (Figure 4H, arrow pointing). Local lymphocytic infiltration is seen within the tumour (Figure 4I, arrow pointing).

Figure 4J–L show a canine sebaceous adenoma. Under low magnification, the mass is seen as multiple basophilic mass-like structures located subcutaneously and well demarcated from the surrounding tissue (Figure 4J, arrow pointing). The interior of the lobules consists of proliferating sebocytes with abundant light-stained eosinophilic vacuolated cytoplasm and centrally located deeply stained small nuclei, with occasional mitosis (Figure 4K, arrow pointing). Infiltration of inflammatory cells is seen in close proximity of the mass to the skin (Figure 4L, arrow pointing).

### 3.6. The Expression of VEGF, SF, P53 and NLRP3 in Tissues

#### 3.6.1. VEGF, SF, P53, NLRP3 Agarose Gel Electropherograms

The results of gel electropherograms of the target genes and reference genes show that clear single amplified bands are obtained at the expected size positions, as shown in Figure 5, and all of them are verified to be the target gene sequences by sequencing.

#### 3.6.2. RT-qPCR Results

As can be seen in Figure 6 and Table 5, the relative expression levels of the marker VEGF mRNA in canine mammary carcinoma are significantly higher than those in the canine benign mammary tumours and the paraneoplastic controls, and the differences are highly significant (*p* < 0.01), and the differences between the benign mammary tumours and the paraneoplastic controls are significant (*p* < 0.05). The relative expression levels of the tumour marker P53 mRNA in the canine mammary carcinomas are highly significant (*p* < 0.001) when compared with those in the benign mammary tumours and the paraneoplastic controls, but the differences between the two are not significant (*p* > 0.05) when the benign mammary tumours are compared with the paraneoplastic controls. The relative expression levels of SF mRNA in the canine mammary carcinomas are highly significant comparing with those in the benign mammary tumours and the paraneoplastic controls (*p* < 0.01), and the differences between the benign mammary tumours and the paraneoplastic controls are significant comparing with those in the benign mammary tumours (*p* < 0.05). The relative expression levels of the tumour marker NLRP3 mRNA in the canine mammary carcinomas are highly significant (*p* < 0.01) when compared with those in the benign mammary tumours and the paraneoplastic controls, and the differences between the benign mammary group and the paraneoplastic control group are not significant (*p* > 0.05).

### 3.7. The Expression of VEGF, SF, P53 and NLRP3 in Sera

#### 3.7.1. ELISA Results

As shown in Table 6 and Figure 7, the expression level of the tumour marker VEGF in the serum of dogs with canine mammary carcinomas is significantly higher than that in the group with benign mammary tumours and the healthy controls (*p* < 0.001), and the differences between the group with benign mammary tumours and the healthy control is significant (*p* < 0.05). The expression level of the tumour marker P53 in the serum of dogs with canine mammary carcinomas is significantly higher than that in the group with benign mammary tumours and the healthy control (*p* < 0.001), but the differences are not significant when comparing the values of the benign mammary tumours and the healthy control groups (*p* > 0.05). The differences among the expression levels of tumour marker SF in the serum of dogs with canine mammary carcinomas, benign mammary tumours and healthy controls are highly significant (*p* < 0.001), and the differences between the benign mammary tumours and healthy controls are significant (*p* < 0.01). The expression level of NLRP3 in the serum of dogs with canine mammary carcinomas is significantly higher than that of dogs with benign tumours and healthy controls (*p* < 0.001), and the differences between the benign mammary tumour and the healthy control groups are not significant (*p* > 0.05).

#### 3.7.2. Assessment of the Value of VEGF, SF, P53, and NLRP3 as Individual Tests for Canine Mammary Carcinoma

According to the results in Table 7, the sensitivity of each single test for each tumour marker in canine mammary carcinoma is P53 (63.3%) > SF (56.7%) > VEGF (53.3%) > NLRP3 (50.0%) in descending order; taking the benign mammary tumour group as the control, the specificity of the four markers is VEGF (85.2%) > P53 (81.5%) > SF (77.8%) > NLRP3 (70.4%); accuracy is P53 (71.9%) > VEGF (68.4%) > SF (66.7%) > NLRP3 (59.7%); and the Youden index is P53 (0.448) > VEGF (0.385) > SF (0.345) > NLRP3 (0.204).

In summary, the serum tumour marker with the highest sensitivity and accuracy is P53, at 63.3% and 71.9%, respectively, and the serum tumour marker with the highest specificity is VEGF, 85.2%. Although all four tumour markers are valuable for the early diagnosis of canine mammary carcinoma, the magnitude of Youden’s index suggests that the tumour markers VEGF and P53 have greater value for the early diagnosis and prognosis of canine mammary carcinoma.

#### 3.7.3. Comparison of the Value of Combined VEGF, SF, P53 and NLRP3 Assays for the Assessment of Canine Mammary Carcinoma

As shown in Table 8, the sensitivity of the two-by-two combined assays in VEGF, P53, SF and NLRP3 is SF+VEGF (66.7%), SF+P53 (73.3%), SF+NLRP3 (63.3%), VEGF+P53 (70.0%), VEGF+NLRP3 (73.3%), and P53+NLRP3 (80.0%). Specificity is SF+VEGF (74.1%), SF+P53 (71.9%), SF+NLRP3 (71.9%), VEGF+P53 (77.8%), VEGF+NLRP3 (66.7%), and P53+NLRP3 (66.7%), respectively. The accuracy is SF+VEGF (70.2%), SF+P53 (72.0%), SF+NLRP3 (66.7%), VEGF+P53 (73.7%), VEGF+NLRP3 (70.2%), and P53+NLRP3 (73.7%). The Youden index, in descending order, was VEGF+P53 > P53+NLRP3 > SF+P53 > SF+VEGF > VEGF+NLRP3 > SF+NLRP3. Taken together, the best detection effect when combining the three tumour markers is achieved with the SF+VEGF+P53 combination. The sensitivity, accuracy and Youden index of the combined detection of VEGF, P53, SF and NLRP3 are as high as 93.3%, 75.4% and 0.489 in turn, but the specificity is the lowest at 55.6%. Comparing with VEGF, P53, SF and NLRP3 alone, the sensitivity, accuracy and Youden index of two-, three- and four-indicator combined tests are elevated to different degrees, which could greatly improve the clinical diagnostic significance. However, the more markers involved in the combined test, the lower the diagnostic specificity, thus making the misdiagnosis rate increase, but the specific clinical application value needed requires further research and discussion.

#### 3.7.4. Area under the Curve for VEGF, SF, P53, and NLRP3 Alone and in Combination

Receiver operating characteristic curves can easily identify the disease recognition ability at any threshold value. Plotting the ROC curve and calculating the area under the curve (AUC) can help to better assess the value of these markers for the diagnosis of canine mammary carcinoma. As known in Table 9 and Figure 8, each of the tumour markers is significant in the diagnosis of canine mammary carcinoma (AUC > 0.5). The highest AUC for a single tumour marker is P53 (AUC = 0.749), followed by VEGF (AUC = 0.736), and the lowest is NLRP3 (AUC = 0.679). When two tumour markers are detected in combination, the highest AUC is for SF+P53 (AUC = 0.827), followed by VEGF+P53 (AUC = 0.807), and the lowest is for SF+NLRP3 (AUC = 0.772). When multiple tumour markers are tested in combination, the highest AUC is for the combination of VEGF, P53, SF and NLRP3 (AUC = 0.879). This means that the diagnostic yield of the combined test is much higher than that of any single test.

## 4. Discussion

Tumours are one of the clinical risk factors that threaten the life and health of dogs [27]. Benign tumours can be controlled or cured by surgery and other treatments, but malignant tumours may be life-threatening, so early diagnosis is particularly important. There are many clinical methods for the diagnosis of malignant disease, including general clinical examination, imaging examination, haematological examination, etc. Different diagnostic methods will be chosen in different conditions. 

Tumour marker examination is a means of clinical diagnosis of tumour diseases, but it has certain limitations, so it is usually combined with imaging examination and puncture biopsy to make a clear diagnosis. A tumour marker is a kind of substance that can reflect the existence of a tumour, being produced by the expression, secretion or apoptosis of the tumour cells themselves, and it mainly exists in the blood, body fluids, tissues and other substances. Usually, tumour markers are used to detect the presence and content of the tumour, so as to screen, diagnose and monitor the efficacy of oncological diseases, etc., which plays a key role in the prevention and treatment of oncological diseases [28].

Four markers, VEGF, NLRP3, P53 and SF, were selected for testing in this experiment. VEGF is a dimeric protein generally produced by tumour cells, macrophages, plasma cells and lymphocytes to stimulate angiogenesis in vitro and in vivo by inducing endothelial cell proliferation and migration. VEGF is overexpressed in many human malignancies and considered to be an important angiogenic factor in human oncology, but the factor remains poorly studied in veterinary oncology [29]. There have been experimental studies using animal models that have shown that inhibition of VEGF production by the administration of anti-VEGF antibodies is associated with a reduction in tumour growth [30]. 

The NLRP3 inflammatory vesicle plays a key role in different types of cancers and is the most comprehensively studied inflammatory vesicle involved in the development of cancer. Studies have shown that activation of NLRP3 inflammatory vesicles induces aberrant secretion of soluble cytokines, generates a favourable inflammatory environment to support tumour growth, and further promotes tumourigenesis and progression by inducing oxidative damage to the DNA of histiocytes, which leads to uncontrolled proliferation [31]. 

SF is one of the most iron-rich proteins in the body, and in human medicine, SF in serum is an important indicator for determining the body’s iron storage capacity, which is significant in the diagnosis of iron-deficiency anaemia, overloading of iron, and the investigation of nutritional status. Meanwhile, SF, as a tumour marker, is related to cell proliferation and has a certain reference value for the diagnosis of certain malignant tumours in clinical practice [32]. When the level of SF increases in serum, it may be caused by an increase of ferritin synthesised by cancer cells [33]. Because canine mammary tumour is a good research model for human breast cancer, SF was selected as one of the markers to be detected in this experiment to further observe the expression of this factor in canine mammary tumours. 

P53 is a tumour suppressor gene that initiates the translation of proteins in response to DNA damage or activation of oncogenes, inducing cell cycle arrest, apoptosis, senescence, DNA repair or metabolic changes. Mutations in this gene occur in more than 50% of all malignant tumours [34]. The P53 protein transforms after mutation due to a change in its spatial conformation, causing the loss of its regulatory role in cell growth, apoptosis and DNA repair [35]. 

In previous studies, progesterone receptor, human epidermal growth factor receptor-2 and oestrogen receptor have been widely studied in the diagnosis of canine mammary carcinoma [36]. As we all know, the concentrations of these three markers show abnormal levels in mammary carcinoma, so they are commonly used in the detection of this disease [37]. But in order to explore new detection combinations, we screened four biomarkers based on the mechanism of detecting tumours, benign tumour tissues and malignant tumour tissues, detected by RT-qPCR, and the results showed that all four markers were significantly expressed in malignant tumours, which was consistent with the findings in the existing relevant literature. The ELISA results proved that the expression trends of the four markers in tissues and serum were consistent, and the characteristics of the four markers in tumour diagnosis were further analysed in this experiment, both individually and in combination. The results showed that the combined detection of the four markers had the highest accuracy and sensitivity, but the disadvantage was that the specificity would be reduced, so the best tumour marker for a certain type of mammary tumour needs further testing and screening. However, the combined detection of tumour markers may provide some reference and help for the development of tumour detection kits in clinical practice.

## 5. Conclusions

Since canine tumour is a common canine clinical disease, there are many ways to diagnose the disease clinically. The clinical diagnostic methods used in this experiment included general clinical examinations such as palpation and visual examination, hematological examinations such as blood routine and biochemistry, imaging examinations such as ultrasound and X-ray, cytological examinations, and histopathological diagnosis, while the detection of tumour markers had landmark significance in tumour diagnosis. The relative expression of four tumor markers VEGF, P53, SF and NLRP3 in canine mammary tumor tissue and paracancerous tissue was measured by RT-qPCR, a molecular biology method. It was found that the four tumor markers were highly expressed in canine mammary carcinoma, which was significantly different from the benign group and control group (*p* < 0.01). The content of the four biomarkers in serum was detected using ELISA, and the expression trend was consistent with that in tissues. It was found that the combined detection of these four biomarkers improved the accuracy and sensitivity of canine breast tumor diagnosis, but the specificity decreased. We hope that this combined detection method can make a great contribution to the early diagnosis of tumor related diseases in dogs.

## Figures and Tables

**Figure 1 animals-14-01272-f001:**
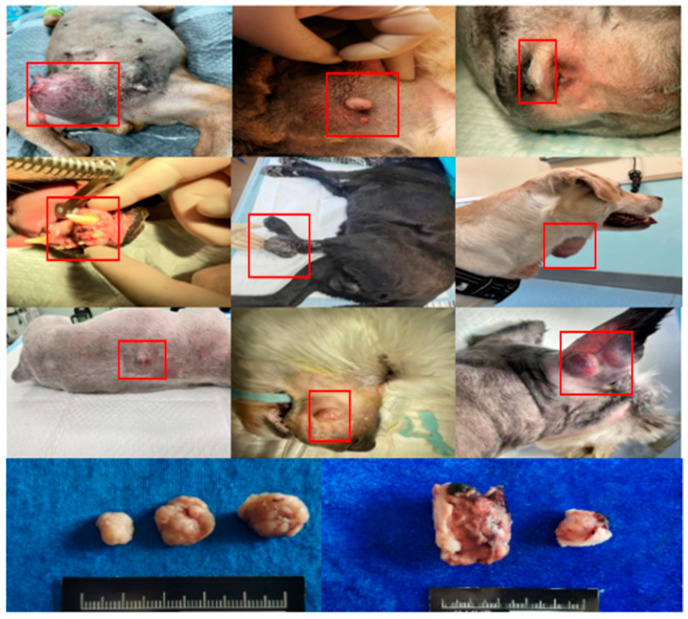
Pictures of clinical signs in dogs with tumours.

**Figure 2 animals-14-01272-f002:**
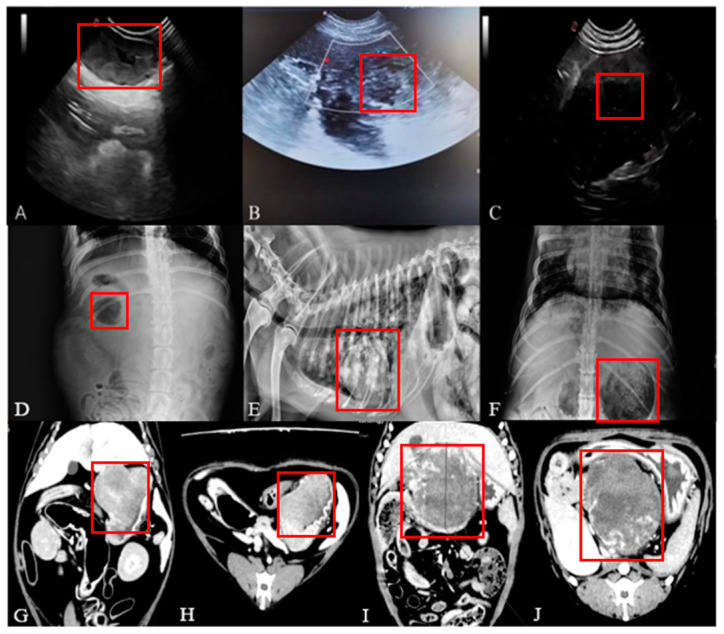
Canine tumour imaging results.

**Figure 3 animals-14-01272-f003:**
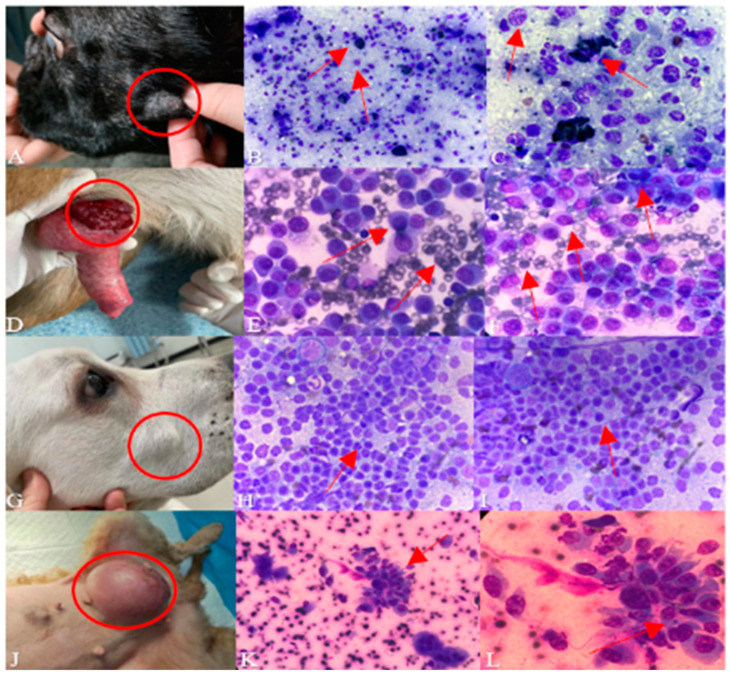
Cytological results in canine tumours.

**Figure 4 animals-14-01272-f004:**
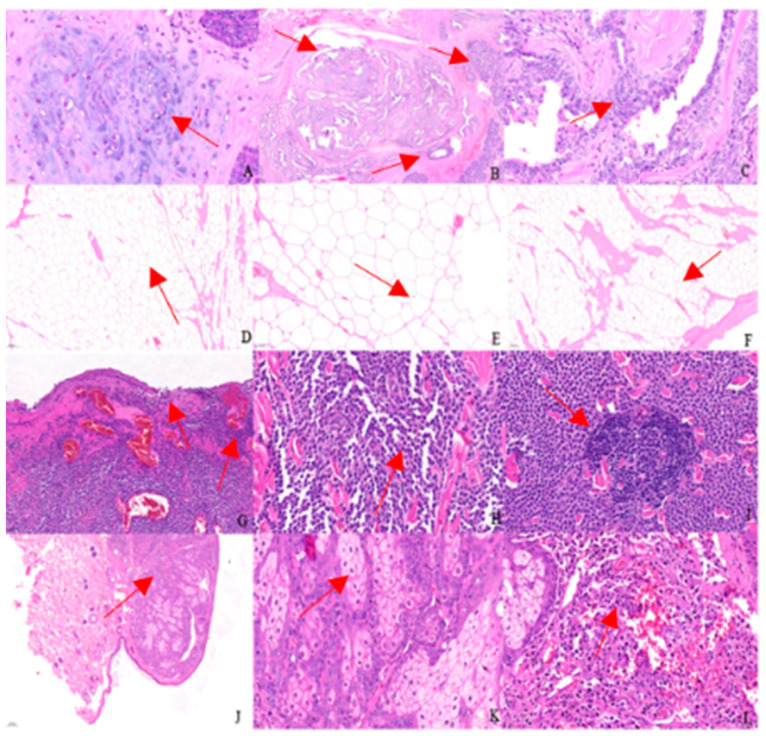
(**A**–**L**) show the histopathological results in canine tumours.

**Figure 5 animals-14-01272-f005:**
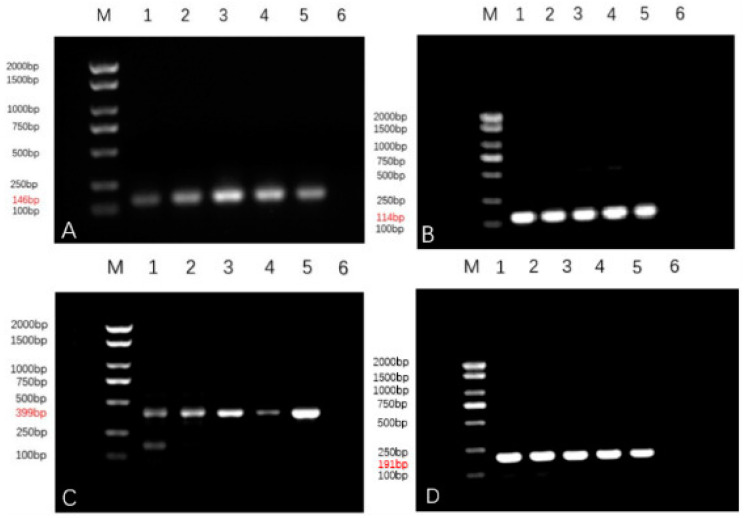
(**A**–**D**) show the gel electrophoresis results of SF, NLRP3, VEGF and P53.

**Figure 6 animals-14-01272-f006:**
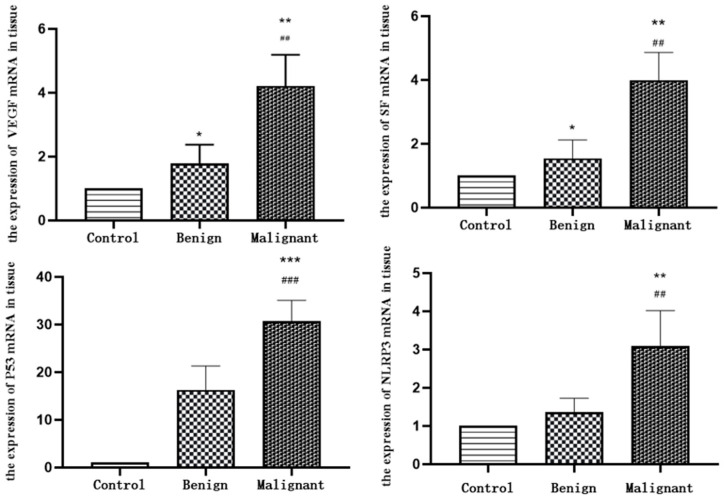
The relative expression levels of markers NLRP3, P53, SF and VEGF mRNA. ① * represents a significant difference (*p* < 0.05); ** represents a highly significant difference (*p* < 0.01), and *** represents a highly significant difference (*p* < 0.0001). ② ## represents a highly significant difference (*p* < 0.01), and ### represents a highly significant difference (*p* < 0.0001).

**Figure 7 animals-14-01272-f007:**
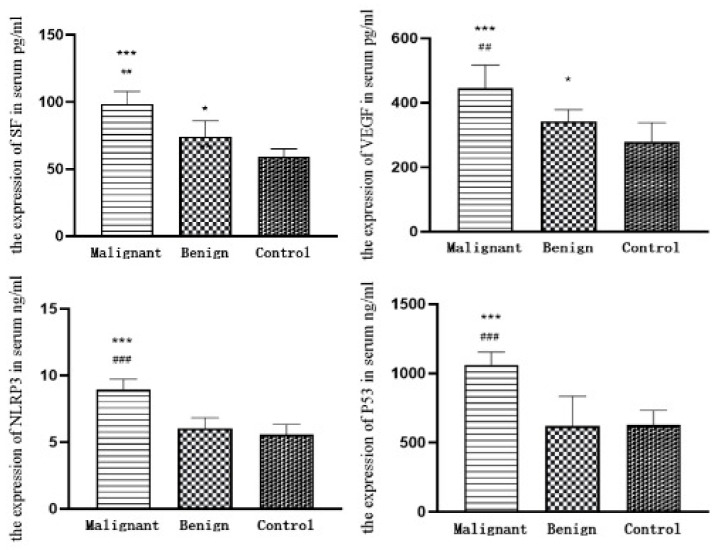
The results of SF, P53, VEGF and NLRP3 determination in the serum of the three groups. ① * represents a significant difference (*p* < 0.05); ** represents a highly significant difference (*p* < 0.01), and *** represents a highly significant difference (*p* < 0.0001). ② ## represents a highly significant difference (*p* < 0.01), and ### represents a highly significant difference (*p* < 0.0001).

**Figure 8 animals-14-01272-f008:**
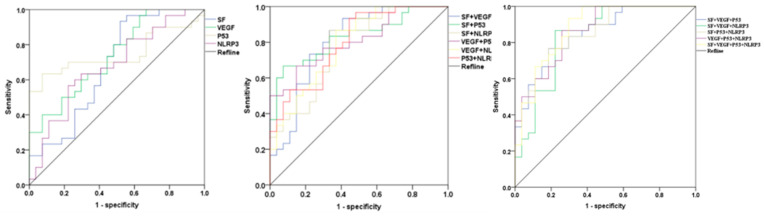
ROC curves of single and combined tumor markers for canine mammary tumor diagnosis.

**Table 1 animals-14-01272-t001:** Gene primer sequence.

Name of Primer	Expected Product/Bp	Sequence (5′–3′)	
SF	146	Sense	GATGCTGCTTCTGGTATGTCCTATCTC
		Anti-sense	GAATACACTCCACCATCCTCTTGACG
VEGF	399	Sense	CAGGCGTATGCAGGCAAAGA
		Anti-sense	GAGGTGGCTTGTGCTGGTGT
P53	191	Sense	GACACAGTGTGGTGGTGCCTTA
		Anti-sense	GGCACAAACGCGTACCTCAA
NLRP3	114	Sense	GCAACAGTGTGAGGTGAGGCTAC
		Anti-sense	TGCAATGCTCTTGGAGACACAGG

**Table 2 animals-14-01272-t002:** Comparison of clinical features.

Group	Canine Number	Mean Age	Tumor Size (Mean ± SD)	Sex	Mammary Gland	Mental State
Healthy dogs	30	4.5	0	female	Smooth	Normal
Benign tumor dogs	179	6.7	1.56 ± 0.48	female	Protrusion	Normal
Malignant tumor dogs	269	8.9	1.82 ± 0.24	female	Redness, swelling, rupture	Bad

**Table 3 animals-14-01272-t003:** Routine blood tests results.

Parameters	Unit	Min–Max	Value
White blood cells (WBC)	10^9^/L	6–17	12.51
Neutrophil percentage (NE%)	%	52.0–81	72.3
Lymphocyte percentage (Lym%)	%	12–33	16.4
Monocyte percentage (Mon%)	%	2.0–13	7.8
Eosinophil percentage (Eos%)	%	1.0–4.0	3.4
Basophil percentage (Bas%)	%	0–1.3	0.1
Basophil (Bas#)	10^9^/L	20–27	21.5
Red blood cells (RBC)	10^12^/L	5.1–8.5	6.04
Hemoglobin (HGB)	g/L	85–153	130
Red blood cell specific volume (HCT)	%	26–47	37.5
Mean corpuscular volume (MCV)	fL	60–76	62
Mean corpuscular hemoglobin (MCH)	pg	20–27	21.5
Mean corpuscular hemoglobin concentration (MCHC)	g/L	300–380	347
Platelet count (PLT)	10^9^/L	117–490	636
Neutrophil (NE#)	10^9^/L	3.62–13.3	9.05
Lymphocyte (Lym#)	10^9^/L	0.83–4.91	2.06
Monocyte (Mon#)	10^9^/L	0.14–1.97	0.97
Eosinophil (Eos#)	10^9^/L	0.04–1.62	0.42

**Table 4 animals-14-01272-t004:** Blood biochemistry results.

Parameters	Unit	Min-Max	Value
Alanine transaminase (ALT)	U/L	5-125	55
Glutamic acid (GLU)	mmol/L	3.89-7.94	5.38
UREA	mmol/L	2.5-9.6	13.64
Creatinine (CREA)	umol/L	44-159	120.3
Total protein (TP)	g/L	52-82	61.7
BUN/CRE		16-218	113.366
Albumin (ALB)	g/L	23-40	29.9
Globulin (GLO)	g/L	25-45	31.8
A/G		0.8-2.0	0.94
Alkaline phosphatase (ALP)	U/L	23-213	411

**Table 5 animals-14-01272-t005:** The results of relative expression of NLRP3, P53, SF and VEGF mRNA in mammary tumours.

Group	NLRP3	P53	SF	VEGF
Control	1	1	1	1
Benign	1.358 ± 0.3664	16.25 ± 5.023	1.541 ± 0.5786 *	1.783 ± 0.5894 *
Malignant	3.086 ± 0.9313 **##	38.70 ± 17.04 ***###	3.989 ± 0.8782 **##	3.932 ± 1.320 **##

① * represents a significant difference (*p* < 0.05); ** represents a highly significant difference (*p* < 0.01), and *** represents a highly significant difference (*p* < 0.0001). ② ## represents a highly significant difference (*p* < 0.01), and ### represents a highly significant difference (*p* < 0.0001).

**Table 6 animals-14-01272-t006:** Comparison of serum SF, P53, VEGF and NLRP3 levels in each group.

Group	SF (ng/mL)	P53 (pg/mL)	VEGF (pg/mL)	NLRP3 (ng/mL)
Malignant	105.39 ± 3.81 ***##	1027.24 ± 54.29 ***###	460.31 ± 19.58 ***##	8.26 ± 0.29 ***###
Benign	88.40 ± 10.11 *	935.45 ± 25.65 *	396.54 ± 25.65 *	7.37 ± 0.47
Control	56.27 ± 5.72	606.82 ± 44.51	246.85 ± 44.51	5.24 ± 0.51

① * represents a significant difference (*p* < 0.05), and *** represents a highly significant difference (*p* < 0.0001). ② ## represents a highly significant difference (*p* < 0.01), and ### represents a highly significant difference (*p* < 0.0001).

**Table 7 animals-14-01272-t007:** Sensitivity, specificity, accuracy and Youden index of single detection of four serum tumor markers.

Marker	Sensitivity (%)	Specificity (%)	Accuracy (%)	Youden Index (%)
SF	56.7 (17/30)	77.8 (21/27)	66.7 (38/57)	0.345
VEGF	53.3 (16/30)	85.2 (23/27)	68.4 (39/57)	0.385
P53	63.3 (19/30)	81.5 (22/27)	71.9 (41/57)	0.448
NLRP3	50.0 (15/30)	70.4 (19/27)	59.7 (34/57)	0.204

**Table 8 animals-14-01272-t008:** Sensitivity, specificity, accuracy and Youden index of joint detections of serum markers SF, VEGF, P53 and NLRP3 levels.

Marker	Sensitivity (%)	Specificity (%)	Accuracy (%)	Youden Index
SF+VEGF	66.7 (20/30)	74.1 (20/27)	70.2 (40/57)	0.408
SF+P53	73.3 (22/30)	71.9 (19/27)	72.0 (41/57)	0.452
SF+NLRP3	63.3 (19/30)	71.9 (19/27)	66.7 (38/57)	0.352
VEGF+P53	70.0 (21/30)	77.8 (21/27)	73.7 (42/57)	0.478
VEGF+NLRP3	73.3 (22/30)	66.7 (18/27)	70.2 (40/57)	0.400
P53+NLRP3	80.0 (24/30)	66.7 (18/27)	73.7 (42/57)	0.467
SF+VEGF+P53	83.3 (25/30)	63.0 (17/27)	73.7 (42/57)	0.463
SF+VEGF+NLRP3	76.7 (23/30)	62.9 (17/27)	70.2 (40/57)	0.396
SF+P53+NLRP3	83.3 (25/30)	59.3 (16/27)	71.9 (41/57)	0.426
VEGF+P53+NLRP3	86.7 (26/30)	59.3 (16/27)	73.7 (42/57)	0.460
SF+VEGF+P53+NLRP3	93.3 (28/30)	55.6 (15/27)	75.4 (43/57)	0.489

**Table 9 animals-14-01272-t009:** The area under the ROC curve of tumor markers VEGF, NLRP3, P53 and SF.

Marker	AUC	*p*	95%CI
SF	0.683	0.018	0.540~0.826
VEGF	0.736	0.002	0.608~0.863
P53	0.749	0.001	0.615~0.884
NLRP3	0.679	0.020	0.539~0.819
SF+VEGF	0.804	<0.001	0.686~0.921
SF+P53	0.827	<0.001	0.720~0.934
SF+NLRP3	0.772	<0.001	0.649~0.894
VEGF+P53	0.807	<0.001	0.697~0.918
VEGF+NLRP3	0.789	<0.001	0.672~0.906
P53+NLRP3	0.796	<0.001	0.682~0.910
SF+VEGF+P53	0.852	<0.001	0.756~0.948
SF+VEGF+NLRP3	0.833	<0.001	0.724~0.942
SF+P53+NLRP3	0.852	<0.001	0.756~0.947
VEGF+P53+NLRP3	0.863	<0.001	0.771~0.955
SF+VEGF+P53+NLRP3	0.879	<0.001	0.790~0.968

## Data Availability

The data that support the findings of this study are available from the authors, upon reasonable request.

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
