# Peer review of "Diagnosis of Canine Tumours and the Value of Combined Detection of VEGF, P53, SF and NLRP3 for the Early Diagnosis of Canine Mammary Carcinoma"

_animals, 2024, doi:10.3390/ani14091272_

Round 1
Reviewer 1 Report
Comments and Suggestions for Authors
Dear Authors,
Firstly, congratulations on your work.
I have several queries regarding the content presented in your paper.
Firstly, you mentioned a total of 448 cases, yet only 30 were identified as malignant tumors. Could you please provide clarification on this apparent discrepancy?
Additionally, I would like to inquire about the criteria used to define a healthy animal in your study. Understanding these conditions would help contextualize the findings and interpretations.
Moreover, I suggest including a new Table 1 to showcase the clinical characteristics, including the number of animals, for both healthy and diseased dogs. This would enhance the comprehensibility of the data presented.
Regarding Table 3, while you've detailed the results for one dog, I'm curious about the outcomes for the other animals studied. Could you please elaborate on this?
In Figure 7, I recommend maintaining the order of control, benign, and malignant as presented in Figure 6 for consistency and clarity.
Concerning Figure 8, I question its inclusion and suggest reconsidering its relevance to the study.
Your conclusion regarding the combined detection of the four markers exhibiting the highest accuracy and sensitivity is intriguing. Could you please elaborate on the advantages of using these four markers compared to solely relying on progesterone receptor, estrogen receptor, and HER2 receptor?
Finally, I believe there is a lack of essential information about the animals and tumors, such as age, lymph node status, metastasis, TNM classification, malignancy grade, tumor necrosis, and molecular subtype. Integrating this data would provide a more comprehensive understanding of the study outcomes.
Thank you for considering my feedback.
Author Response
Reviewer #1:
Comment 1: You mentioned a total of 448 cases, yet only 30 were identified as malignant tumors. Could you please provide clarification on this apparent discrepancy?
Response: We thank the reviewer for the positive comments. We have made modifications to the content by adding description in the abstract.
Comment 2: I would like to inquire about the criteria used to define a healthy animal in your study. Understanding these conditions would help contextualize the findings and interpretations.
Response: We thank the reviewer for the positive comments. Healthy animals in the text refer to dogs that are in good physical condition and free from any diseases and we have also added the description in the manuscript.
Comment 3: I suggest including a new Table 1 to showcase the clinical characteristics, including the number of animals, for both healthy and diseased dogs. This would enhance the comprehensibility of the data presented.
Response: We thank the reviewer for the positive comments. New additions have been made in the manuscript.
Comment 4: Regarding Table 3, while you've detailed the results for one dog, I'm curious about the outcomes for the other animals studied. Could you please elaborate on this?
Response: We thank the reviewer for the positive comments. The relevant description has been modified in the article.
Comment 5: In Figure 7, I recommend maintaining the order of control, benign, and malignant as presented in Figure 6 for consistency and clarity.
Response: We thank the reviewer for the positive comments. The positions of the landmarks in the two figures have been unified.
Comment 6: Concerning Figure 8, I question its inclusion and suggest reconsidering its relevance to the study.
Response: We thank the reviewer for the positive comments. After careful consideration, we have deleted the Figure 8.
Comment 7: Your conclusion regarding the combined detection of the four markers exhibiting the highest accuracy and sensitivity is intriguing. Could you please elaborate on the advantages of using these four markers compared to solely relying on progesterone receptor, estrogen receptor, and HER2 receptor?
Response: We thank the reviewer for the positive comments. Additional description has been provided in the manuscript.
Comment 8: I believe there is a lack of essential information about the animals and tumors, such as age, lymph node status, metastasis, TNM classification, malignancy grade, tumor necrosis, and molecular subtype. Integrating this data would provide a more comprehensive understanding of the study outcomes.
Response: We thank the reviewer for the positive comments. Additional description has been provided in the manuscript.
Reviewer 2 Report
Comments and Suggestions for Authors
Canine mammary tumors are constantly a major challenge in veterinary medicine, therefore early diagnosis is essential.
The manuscript is well written: however, some changes must be done.
All abbreviations mentioned already in the abstract, such as VEGF, P53, SF – should be explained.
The methods used are described in the form of an instruction, especially in the section 2.7.4. as it is written: “remove”, “add”, “seal” etc. Please rewrite it as it was done, using the passive verb forms.
In the section 3 – expression of markers in sera and in tissue should be indicated by which method its was performed as it is not clear for the reader.
Please write also, which ELISA Kits were used for the study.
The section of conclusions should be extended.
Figure 4 – the letters and the arrows should be explained.
References though are up to date are scare, the list should be extended.
Author Response
Reviewer #2:
Comment 1: All abbreviations mentioned already in the abstract, such as VEGF, P53, SF – should be explained.
Response: We thank the reviewer for the positive comments. Modification has been made in the manuscript.
Comment 2: The methods used are described in the form of an instruction, especially in the section 2.7.4. as it is written: “remove”, “add”, “seal” etc. Please rewrite it as it was done, using the passive verb forms.
Response: We thank the reviewer for the positive comments. Modification has been made in the manuscript.
Comment 3: In the section 3 – expression of markers in sera and in tissue should be indicated by which method its was performed as it is not clear for the reader.
Please write also, which ELISA Kits were used for the study.
Response: We thank the reviewer for the positive comments. Supplement has been made in the manuscript.
Comment 4: The section of conclusions should be extended.
Figure 4 – the letters and the arrows should be explained.
Response: We thank the reviewer for the positive comments. The conclusions have been extended and the letters and the arrows also been explained.
Comment 5: References though are up to date are scare, the list should be extended.
Response: We thank the reviewer for the positive comments. The references have been extended.
Reviewer 3 Report
Comments and Suggestions for Authors
This is an interesting paper that falls well within the scope of “Animals”. The study appears to be well performed with the appropriate methods being used and it is understood that the major finding and novelty of this paper is that these 4 markers together may be useful for early diagnosis of canine mammary carcinoma. However, the manuscript needs revision before it can be accepted (see below) and would benefit from inclusion of some recent citations (see below) as well as some editorial help, including help with English language.
The authors should comment on and possibly include the following citations:-
1) The authors should discuss and cite the paper by
Biomed Res Int. 2016;2016:4917387. doi: 10.1155/2016/4917387. Where VEGF and other proangiogenic factors are described as important tissue biomarkers in both dog and human mammary carcinogenesis.
2) Pimentel et al., 2023, Vet Sci 10(6):387. doi: 10.3390/vetsci10060387. state that despite the high expression of vascular endothelial growth factor (VEGF) and its receptor (VEGFR), there is no correlation with overall survival time so the authors should comment on this increasing in malignant versus benign tumors.
3) Curiously, the authors do not cite their own publication “Mol Biol Rep. 2023 Dec;50(12):10617-10625. doi: 10.1007/s11033-023-08863-x. even though this publication suggests p53 is a good marker for canine mammary carcinoma
Minor Points:
- Line 94 “some researchers” but citation is to a paper in 2010 – please cite the original paper
- Line 103 – the looks like the first time “SF” is used so it should be defined here i.e. the text should read “…studies have found that serum ferritin (SF) has a complex…”
- Line 141 “In order to ensure the accuracy of the results, it is usually recommended to take two times, one in the orthostatic position, the other is in the lateral position” better as “In order to ensure the accuracy of the results, it is usually recommended to perform the procedure twice, one in the orthostatic position, the other is in the lateral position”
- CT needs to be defined where it is first mentioned in the text
- Format and spacing for Table 4, Fig. 6, Fig. 7 etc needs correcting to “…NLRP3,P53, SF and VEGF mRNA in mammary tumors.
- Line 420 - ROC needs to be defined and text should read “area under the curve (AUC)”
- Line 426 should read “Tumours are one of the…”
Comments on the Quality of English Language
Some grammatical mistakes need correction including punctuation
Author Response
Reviewer #3:
Comment 1: The authors should discuss and cite the paper by
Carvalho et al., 2016 Biomed Res Int. 2016;2016:4917387. doi: 10.1155/2016/4917387. Where VEGF and other proangiogenic factors are described as important tissue biomarkers in both dog and human mammary carcinogenesis.
Response: We thank the reviewer for the positive comments. We have added this paper.
Comment 2: Pimentel et al., 2023, Vet Sci 10(6):387. doi: 10.3390/vetsci10060387. state that despite the high expression of vascular endothelial growth factor (VEGF) and its receptor (VEGFR), there is no correlation with overall survival time so the authors should comment on this increasing in malignant versus benign tumors.
Response: We thank the reviewer for the positive comments. We have added this paper.
Comment 3: Curiously, the authors do not cite their own publication “Yang NY, Zheng HH, Yu C, Ye Y, Du CT, Xie GH.Mol Biol Rep. 2023 Dec;50(12):10617-10625. doi: 10.1007/s11033-023-08863-x. even though this publication suggests p53 is a good marker for canine mammary carcinoma.
Response: We thank the reviewer for the positive comments. We have added this paper.
Comment 4: Line 94 “some researchers” but citation is to a paper in 2010 – please cite the original paper
Response: We thank the reviewer for the positive comments. We have changed the content.
Comment 5: Line 103 – the looks like the first time “SF” is used so it should be defined here i.e. the text should read “…studies have found that serum ferritin (SF) has a complex…”
Response: We thank the reviewer for the positive comments and correction, we have modified the problems in the paper.
Comment 6: Line 141 “In order to ensure the accuracy of the results, it is usually recommended to take two times, one in the orthostatic position, the other is in the lateral position” better as “In order to ensure the accuracy of the results, it is usually recommended to perform the procedure twice, one in the orthostatic position, the other is in the lateral position”
Response: We thank the reviewer for the positive comments and correction, we have modified the problems in the paper.
Comment 7: CT needs to be defined where it is first mentioned in the text
Response: We thank the reviewer for the positive comments. We have defined it for its first appearance.
Comment 8: Format and spacing for Table 4, Fig. 6, Fig. 7 etc needs correcting to “…NLRP3, P53, SF and VEGF mRNA in mammary tumors.
Response: We thank the reviewer for the positive comments and corrections.
Comment 9: Line 420 - ROC needs to be defined and text should read “area under the curve (AUC)”
Response: We thank the reviewer for the positive comments. We have defined it and modified the mistake.
Comment 10: Line 426 should read “Tumours are one of the…”
Response: We thank the reviewer for the positive comments and correction, we have modified it.
Round 2
Reviewer 1 Report
Comments and Suggestions for Authors The authors did not clarify why they used 30 samples when they had a much higher initial n.Author Response
We have two reasons for choosing the sample size.
Firstly, collecting information on 448 cases of canine tumors is to summarize the commonly used diagnostic methods for canine tumor diseases in clinical practice. During the collection process, not all cases have fresh tumor tissue for experimentation. Therefore, through comparison and screening, the freshest and most suitable 30 benign tumors, 30 malignant tumors, and 30 control groups were selected for the experiment.
Secondly, in the literature we reviewed, it was found that having more than 20 clinical samples can indicate the problem and prove the conclusion (such as: doi: 10.1111/j.1751-0813.2008.00312. x.).
In summary, we chose a sample size of 30 in each group.